# MOTION INVERSION FOR VIDEO CUSTOMIZATION

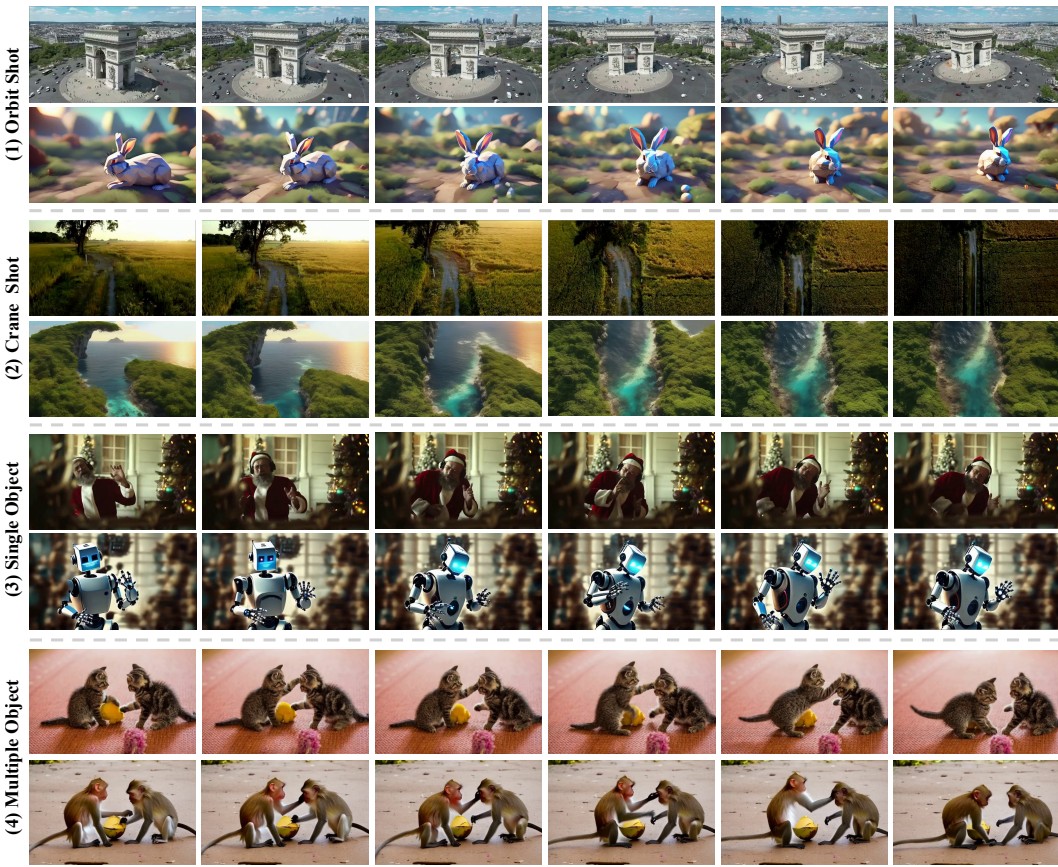

Figure 1: **Applications of the proposed Motion Embeddings for customized video generation**. Our method supports a wide range of motion types, including various camera movements and object motions. In each example, the first row shows the source video, while the second row shows the output. Please refer to the supplementary videos for clearer visualization.

## ABSTRACT

In this work, we present a novel approach for motion customization in video generation, addressing the widespread gap in the exploration of motion representation within video generative models. Recognizing the unique challenges posed by the spatiotemporal nature of video, our method introduces **Motion Embeddings**, a set of explicit, temporally coherent embeddings derived from a given video. These embeddings are designed to integrate seamlessly with the temporal transformer modules of video diffusion models, modulating self-attention computations across frames without compromising spatial integrity. Our approach provides a compact and efficient solution to motion representation, utilizing two types of embeddings: a Motion Query-Key Embedding to modulate the temporal attention map and a Motion Value Embedding to modulate the attention values. Additionally, we introduce an inference strategy that excludes spatial dimensions from the Motion Query-Key Embedding and applies a debias operation to the Motion Value Em-

bedding, both designed to debias appearance and ensure the embeddings focus solely on motion. Our contributions include the introduction of a tailored motion embedding for customization tasks and a demonstration of the practical advantages and effectiveness of our method through extensive experiments.

# 1 INTRODUCTION

In recent years, generative models have rapidly evolved, achieving remarkable results across various domains such as image (Rombach et al., 2022; Nichol et al., 2021; Ramesh et al., 2022; Betker et al., 2023; Saharia et al., 2022) and video (He et al., 2022; Chen et al., 2023a; Guo et al., 2023; Wang et al., 2023). Within the realm of imagery, customization is a popular topic, empowering models to learn specific visual concepts from user-provided images at both the object and style levels. These concepts are combined with the model's extensive prior knowledge to produce diverse and customized outcomes. The success of image customization has led to high expectations for extending such capabilities to video generation models, which are developing rapidly and drawing significant attention.

However, extending these techniques to Text-to-Video (T2V) generation introduces new challenges due to the spatiotemporal nature of video. Unlike images, videos contain motion in addition to appearance, making it essential to account for both. Current customization methods (Hu et al., 2021; Mou et al., 2023; Sohn et al., 2023; Ye et al., 2023; Zhang & Agrawala, 2023; Gal et al., 2022; Ruiz et al., 2023) primarily focus on appearance customization, neglecting motion, which is critical in video. Motion customization deals with adapting specific movements or animations to different objects or characters, a task complicated by the diverse shapes and dynamic changes over time (Siarohin et al., 2019a;b; Yatim et al., 2023; Jeong et al., 2023). These methods, however, fail to capture the dynamics of motion. For instance, textual inversion (Gal et al., 2022) learns embeddings from images but lacks the ability to capture temporal correlations, which are essential for video dynamics. Similarly, fine-tuning approaches like DreamBooth (Ruiz et al., 2023) and LoRA (Hu et al., 2021) struggle to disentangle motion from appearance.

In this paper, we address the challenge of motion customization, focusing on the critical issue of motion representation. The current state-of-the-art methods face several limitations: 1) Some approaches lack a clear representation of motion, as seen in Yatim et al. (2023), where motion is only indirectly injected through loss construction and optimization at test time, leading to additional computational overhead. 2) Some other methods (Jeong et al., 2023) attempt to parameterize motion as a learnable representation, yet they fail to separate these parameters from the generative model. This coupling compromises the generative model's diversity after learning. 3) While there are also some methods that attempt to separate motion representation from the generative model using techniques like low-rank adaptation (LoRA) (Hu et al., 2021), such as in Motion Director (Zhao et al., 2023), they lack a well-defined temporal design, limiting their effectiveness in capturing motion dynamics, as evidenced by our experiments.

To address the aforementioned issues, we propose a novel framework for motion customization. Our method learns explicit, temporally coherent embeddings, termed **Motion Embeddings**, from a reference video. These embeddings are integrated into the temporal transformer modules of the video diffusion model, modulating the self-attention across frames.

We introduce two types of motion embeddings: **Motion Query-Key Embedding**, which captures global relationships between frames by influencing the temporal attention map (QK), and **Motion Value Embedding**, which captures spatially varying movements across frames by modulating the attention value (V). The Motion Query-Key Embedding excludes spatial dimensions ($H$ and $W$) to avoid capturing appearance information, as the temporal attention map inherently carries spatial details of objects, which could entangle appearance information of the reference video and thus hinder motion transfer. While the Spatial-2D Motion Value Embedding may still risk capturing static appearance information, we address this by introducing a debiasing strategy that models frame-to-frame changes, ensuring that the embeddings primarily represent motion dynamics. This approach is conceptually similar to techniques like optical flow, where motion is isolated by tracking changes between frames, helping to prevent overfitting to specific appearance details and improving generalization to novel content.

In summary, our contributions are as follows:

- We propose a novel motion representation for video generation, addressing the key challenges in motion customization.
- We design two approaches to debias appearance for this motion representation: a 1D Motion Query-Key Embedding that captures global temporal relationships, and a 2D Motion Value Embedding with a debias operation that captures spatially varying movements across frames.
- Our method is validated through extensive experiments, demonstrating its effectiveness and flexibility for integration with existing Text-to-Video frameworks.

## 2 RELATED WORK

**Text-to-Video Generation.** Text-to-Video (T2V) generation task aims to synthesize high-quality video from user-provided text prompts, which are composed of the expected appearances and motions. Previously, Generative Adversarial Networks (GANs) (Vondrick et al., 2016; Saito et al., 2017; Pan et al., 2017; Li et al., 2018; Tian et al., 2021) and Autoregressive Transformers (Yan et al., 2021; Le Moing et al., 2021; Wu et al., 2022; Hong et al., 2022) have been widely explored in this area. On the other hand, diffusion models have demonstrated powerful generation capabilities in the field of Text-to-Image (T2I) generation (Rombach et al., 2022; Nichol et al., 2021; Ramesh et al., 2022; Betker et al., 2023; Saharia et al., 2022) and have begun to extend to video generation (He et al., 2022; Chen et al., 2023a; Wang et al., 2023; He et al., 2022). Recently, several works have tried to transfer the pretrained T2I diffusion models to T2V generation models by inserting temporal layers into the image generation networks such as AnimateDiff (Guo et al., 2023), and Make-a-Video (Singer et al., 2022). These Text-to-Video (T2V) models approach frame generation as a series of image creations, integrating a temporal transformer to bolster inter-frame relationships—a notable deviation from Text-to-Image (T2I) models (He et al., 2022; Chen et al., 2023a; Wang et al., 2023; Singer et al., 2022; Zhang et al., 2023; 2024; Chen et al., 2024; cerspense, 2023). Additionally, certain approaches incorporate an extra 3D convolutional layer to enhance temporal consistency (cerspense, 2023; Wang et al., 2023). These T2V models are designed for video generation through text inputs and may encounter difficulties when needed to produce videos with customized motions.

**Video Editing.** Video editing generates video that adheres to the target prompt as well as preserves the spatial layout and motion of the input video. As the basis of video editing, image editing has achieved significant progress by manipulating the internal feature representation of prominent T2I diffusion models (Cao et al., 2023; Chefer et al., 2023; Hertz et al., 2022; Ma et al., 2023b; Tumanyan et al., 2023; Patashnik et al., 2023; Bar-Tal et al., 2022; Qi et al., 2023; Liu et al., 2023). MagicEdit (Liew et al., 2023) takes use of SDEdit(Meng et al., 2021) for each video frame to conduct high-fidelity editing. Tune-A-Video (Wu et al., 2023) finetunes a T2I model on the source video and stylizes the video or replaces object categories via the fine-tuned model. Control-A-Video (Chen et al., 2023b) presents Video-ControlNet, a T2V diffusion model that generates high-quality, consistent videos with fine-grained control by incorporating spatial-temporal attention and novel noise initialization for motion coherence. TokenFlow(Geyer et al., 2023) performs frame-consistent editing by the feature replacement between the nearest neighbor of target frames and keyframes. However, these methods fall short as they just duplicate the original motion almost at pixel-level, resulting in failures when being require significant structural deviation from the original video.

**Video Motion Customization.** Motion customization involves generating a video that maintains the motion traits from a source video, such as direction, speed, and pose, while transforming the dynamic object to match a text prompt's specified visual characteristics. This process is distinct from video editing (Bar-Tal et al., 2022; Chen et al., 2023b; Wu et al., 2023; Geyer et al., 2023; Liew et al., 2023; Qi et al., 2023), which typically transfers motion between similar videos within the same object category. In contrast, motion customization requires transferring motion across diverse object categories, often involving significant shape and deformation changes over time, necessitating a deep understanding of object appearance, dynamics, and scene interaction (Yatim et al., 2023; Jeong et al., 2023; Zhao et al., 2023; Ling et al., 2024; Jeong et al., 2024). Diffusion Motion Transfer (DMT) (Yatim et al., 2023) injects the motion of the reference video through the guidance of a

handcrafted loss during inference, bringing additional computation costs that could not be ignored. Video Motion Customization (VMC) (Jeong et al., 2023) encodes the motion into the parameters of a T2V model. However, finetuning the original T2V model could seriously limit the diversity of the generation model after learning the motion. Motion Director(Zhao et al., 2023) adopts LoRA(Hu et al., 2021) to embed the motion outside the T2V model. Nevertheless, the structure of LoRA limits the scalability and interpretability, as we could not easily integrate several reference motions by these methods. Other works that represent motion using parameterization (Wang et al., 2024; He et al., 2024) or trajectories (Ma et al., 2023a; Yin et al., 2023), but these approaches fall outside the scope of our discussion on reference video-based methods.

## 3 METHODOLOGY

### 3.1 TEXT-TO-VIDEO DIFFUSION MODEL

In video diffusion models, the evolution from Text-to-Image (T2I) to Text-to-Video (T2V) models is marked by the introduction of the temporal transformer module to the basic block. While T2V models utilize **spatial convolutional layers** and **spatial transformers** in basic block for integrating image features and textual information (Rombach et al., 2022; Nichol et al., 2021; Ramesh et al., 2022; Betker et al., 2023; Saharia et al., 2022) , T2V models build on this by adding the **temporal transformer** (He et al., 2022; Chen et al., 2023a; Guo et al., 2023; Wang et al., 2023). This module is key for video generation, enabling the treatment of videos as sequences of batched images. It specifically handles the inter-frame correlations through a frame-level self-attention mechanism, ensuring the temporal continuity vital for dynamic video content.

In this module, an input spatiotemporal feature tensor is provided, initially shaped as $\mathbf{X} \in \mathbb{R}^{1 \times C \times N \times H \times W}$, where $C, N, H, W$ represents channels, number of frames, height, and width respectively. Batch size equals to one, and we omit the batch size dimension in our later notation for simplicity. This tensor is subsequently transformed into a feature tensor $\mathbf{F}$, with dimensions $\mathbf{F} \in \mathbb{R}^{(H \times W) \times N \times C}$. The temporal attention mechanism (TA) within this module specifically targets the $N$ dimension, corresponding to frames.

To facilitate this operation, $F$ is projected through three distinct linear layers to generate the Query ($\mathbf{Q} = \mathbf{W_q F}$), Key ($\mathbf{K} = \mathbf{W_k F}$), and Value ($\mathbf{V} = \mathbf{W_v F}$) matrices, respectively. This setup enables the execution of self-attention across the frame sequence, encapsulated by the formula:

$$\mathrm{TA}(\mathbf{F}) = \mathrm{softmax}\left(\frac{\mathbf{Q}\mathbf{K}^{\mathbf{T}}}{\sqrt{d_k}}\right)\mathbf{V}, \tag{1}$$

where $\mathbf{Q}$, $\mathbf{K}$, and $\mathbf{V}$ are the query, key, and value matrices obtained from $\mathbf{F}$, and $d_k$ represents the dimensionality of the key vectors, serving as a scaling factor to maintain numerical stability within the softmax function. This temporal attention mechanism allows each frame's updated feature to gather information from other frames, enhancing the inter-frame relationships and capturing the temporal continuity essential for video generation.

### 3.2 OUR PROPOSED METHOD

At the heart of our method for enhancing inter-frame dynamics in video models is the innovative **motion embedding** concept:

$$\begin{aligned} \mathcal{M} &= \{\mathcal{M}^{\mathcal{QK}}, \mathcal{M}^{\mathcal{V}}\}, \\ \mathcal{M}^{\mathcal{QK}} &= \{\mathbf{m}_1^{QK}, \mathbf{m}_2^{QK}, \ldots, \mathbf{m}_L^{QK}\}, \quad \text{where each } \mathbf{m}_i^{QK} \in \mathbb{R}^{1 \times N \times C}, \\ \mathcal{M}^{\mathcal{V}} &= \{\mathbf{m}_1^V, \mathbf{m}_2^V, \ldots, \mathbf{m}_L^V\}, \quad \text{where each } \mathbf{m}_i^V \in \mathbb{R}^{(H*W) \times N \times C}. \end{aligned} \tag{2}$$

We have designed two distinct types of motion embeddings, each influencing different parts of the temporal attention computation. The **Motion Query-Key Embedding** $\mathbf{m}_i^{QK}$ is a learnable vector with the shape $(1, N, C)$, while **Motion Value Embedding** $\mathbf{m}_i^V$ is a learnable matrix with the shape $(H \times W, N, C)$. These embeddings are seamlessly integrated into the spatiotemporal feature tensor

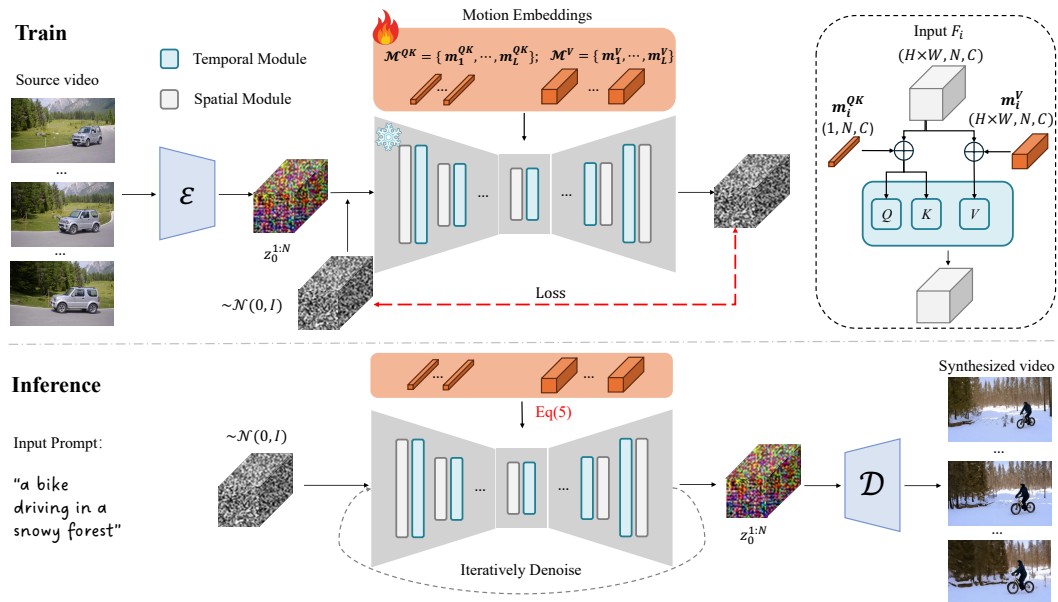

Figure 2: **Motion Inversion within T2V diffusion models**. The **top** depicts the training phase, where motion embeddings $\mathcal{M}$ are learned by backpropagating the loss through the temporal transformer, influencing the spatiotemporal feature tensor $\mathbf{F}$. These embeddings are then used to modify the self-attention computations within the temporal transformer modules, ensuring enhanced interframe dynamics. The **bottom** shows the inference phase, where an input text prompt guides the generation of a coherent video sequence with the learned motion embeddings applied across the frames, producing a customized video output with desired motion attributes.

$\mathbf{F}$. The variable $L$ denotes the total number of temporal attention modules within the model. Our motion embeddings directly influence the self-attention computation as follows:

$$\text{TA}_i(\mathbf{F}) = \text{softmax}\left(\frac{(\mathbf{W_q}(\mathbf{F} + \mathbf{m}_i^{QK}))(\mathbf{W_k}(\mathbf{F} + \mathbf{m}_i^{QK}))^T}{\sqrt{d_k}}\right)(\mathbf{W_v}(\mathbf{F} + \mathbf{m}_i^V)), \tag{3}$$

**Training** Obtaining this embedding is both efficient and convenient. Given a custom video $x_0^{1:N}$, $N$ equals to number of frames of this video, we zero-initialize each motion embedding and train the video diffusion model and backpropagate the gradient to the motion embedding:

$$\mathcal{M}_* = \arg\min_{\mathcal{M}} \mathbb{E}_{t,\epsilon}\left[\left\|\epsilon_t^{1:N} - \epsilon_\theta(x_t^{1:N}, t, \mathcal{M})\right\|_2^2\right], \tag{4}$$

where $\epsilon_t$ represents the noise variable at time step $t$, and $\epsilon_\theta$ denotes the noise prediction from the pre-trained video diffusion model parameterized by $\theta$. The whole process is shown in Figure 3. Our method also supports the loss formulation of (Jeong et al., 2023) and (Zhao et al., 2023), while the latter we found in the experiment can boost our performance too.

**Inference** During inference time, we apply a differencing operation to modify the optimized motion value embedding and debias the appearance:

$$\tilde{\mathbf{m}}_i^V[:,j,:] = \begin{cases} \mathbf{m}_i^V[:,j,:], & j = 1 \\ \mathbf{m}_i^V[:,j,:] - \mathbf{m}_i^V[:,j-1,:], & j > 1 \end{cases} \tag{5}$$

### 3.3 ANALYSIS

The motivation of our approach is to fully capture the motion information from the target video without being influenced by its appearance. In this section, we analyze how $\mathcal{M}^{QK}$ and $\mathcal{M}^V$ is designed to achieve this.

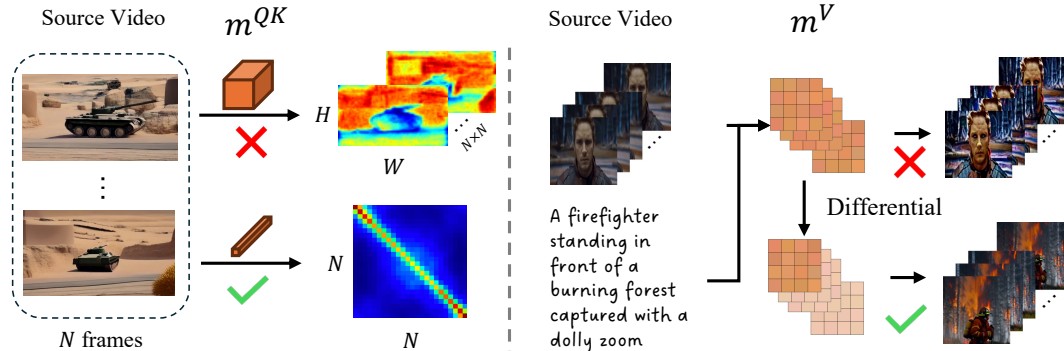

Figure 3: **Debiasing appearance from Motion Embeddings**. **Left**: For the Motion Query-Key Embedding, which influences the attention map, we exclude the spatial dimensions. Including them would cause the attention map between frames to capture the object's shape (e.g., the shape of the tank in the original video is visible in the attention map). **Right**: Following the concept of optical flow, we apply a debias operation to the Spatial-2D Motion Value Embedding, removing static appearance and preserving dynamic motion.

**Motion Query-Key Embedding ($\mathcal{M}^{QK}$)**   The Motion Query-Key Embedding $\mathcal{M}^{QK}$ is designed to influence the attention map within the temporal transformer modules by adjusting the query and key components. By adding $\mathcal{M}^{QK}$ to $\mathbf{F}$ before the projection to queries and keys via Equation 3, we effectively modify the computation of the attention weights. These attention weights determine how frames attend to each other across time, which are critical in modeling the motion of the target video.

Additionally, the shape of $\mathbf{m}_i^{QK} \in \mathbb{R}^{1 \times N \times C}$ is designed to exclude spatial dimensions ($H$ and $W$), which is crucial for removing appearance information. The rationale behind this is that the temporal attention map models the relationships between spatial regions across frames, inherently carrying the appearance information of objects. The temporal attention map has a shape of $(H \times W) \times N \times N$. By examining any one of the attention maps, which has the shape $H \times W$, the object's shape becomes apparent, as illustrated in Figure 3. If $\mathbf{m}_i^{QK}$ included spatial dimensions, appearance details would be captured in the embedding, limiting the ability to transfer motion to new content.

**Motion Value Embedding ($\mathcal{M}^V$)**   As $\mathcal{M}^{QK}$ excludes spatial dimensions, it is better suited for representing global motion (e.g., camera motion) but is less effective at capturing local motion (e.g., instance motion). To address this, we incorporate the Motion Value Embedding $\mathcal{M}^V$ in our representation. Specifically, $\mathbf{m}_i^V \in \mathbb{R}^{(H \times W) \times N \times C}$ includes spatial dimensions, allowing the embedding to represent motion at each spatial location across time frames. This fine-grained representation is essential for modeling local object movements and detailed motion information, enhancing the realism and coherence of the generated videos.

However, $\mathcal{M}^V$ may capture static appearance information, leading to overfitting and limiting generalization. To address this, we apply the differencing operation from Equation 5, which isolates dynamic motion by subtracting the motion value embedding of the previous frame from the current one, removing static appearance. This approach, similar to optical flow, ensures $\mathcal{M}^V$ focuses on motion dynamics, improving generalization to novel text prompts.

## 4 EXPERIMENT

In this section, we employ three **motion customization methods** as our baselines: Diffusion Motion Transfer - CVPR24 (DMT) (Yatim et al., 2023), Video Motion Customization - CVPR24 (VMC) (Jeong et al., 2023), and Motion Director (Zhao et al., 2023). For discussions with **video editing method**, please refer to the supplementary files. To ensure a fair comparison, both our approach and the baseline methods are integrated with the same T2V model, ZeroScope (cerspense, 2023) in all experiments.

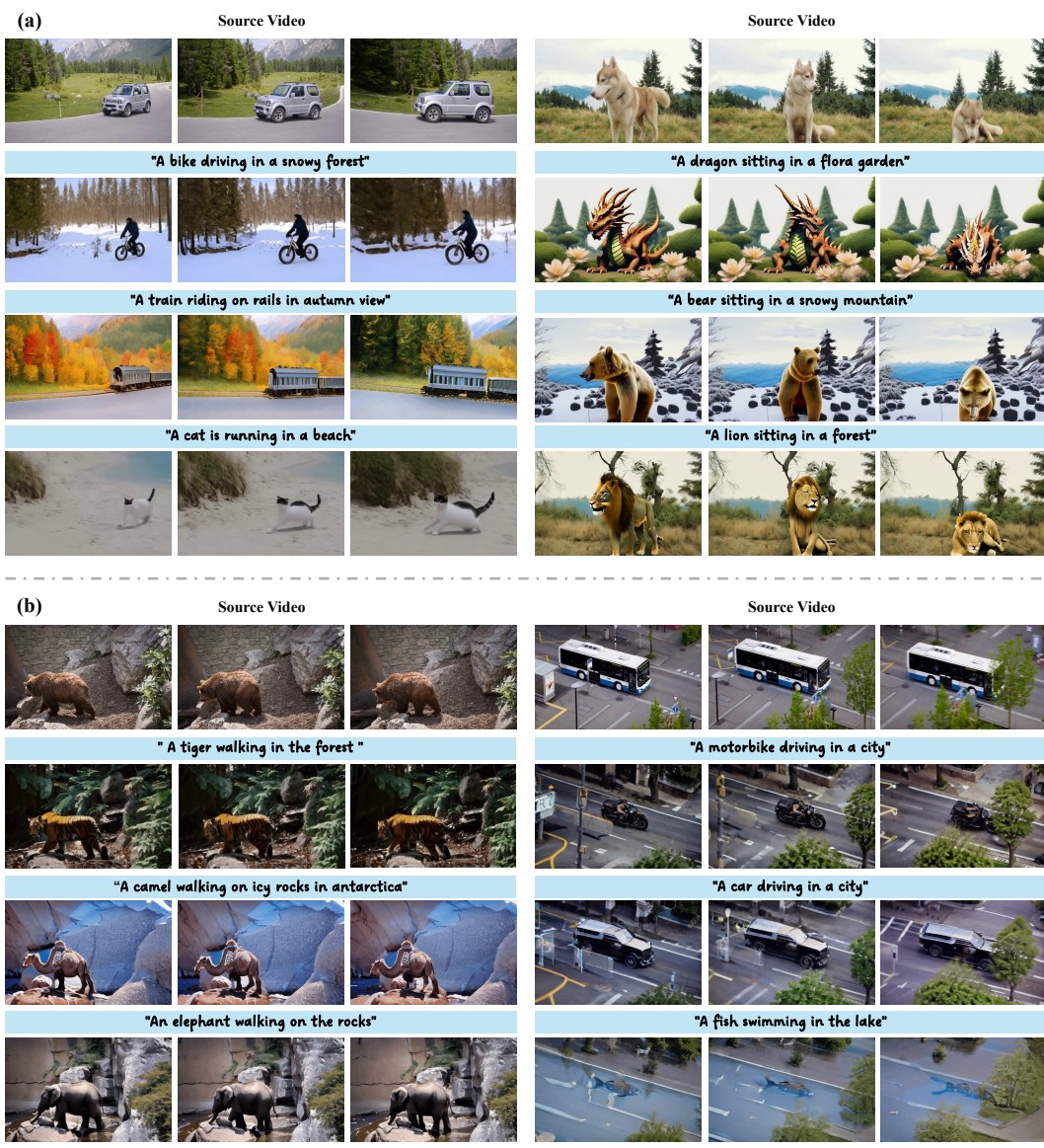

Figure 4: **Sample results of our method.** Our framework demonstrates exceptional adaptability in capturing a broad spectrum of movements, accurately representing everything from subtle gestures to intricate dynamic actions across various source videos. It also exhibits remarkable flexibility in responding to diverse textual prompts, enabling users to guide the synthesis process with descriptive language for customized motion outputs. Furthermore, our method seamlessly integrates with a range of T2V models such as (a) zero-scope (cerspense, 2023) and (b) animate-diff (Guo et al., 2023), showcasing its effectiveness in enhancing video generation with contextually rich and varied motion patterns.

To be consistent with prior work (Yatim et al., 2023; Jeong et al., 2023), our evaluation utilizes source videos from the DAVIS dataset (Perazzi et al., 2016), WebVID (Bain et al., 2021), and various online resources. These videos represent a wide range of scenes and object categories and include a variety of motion types. Comprehensive details on the validation set, prompts used, and implementation specifics of both our method and the baselines are provided in the Supplementary files. Figure 4 showcases examples of our method in action, illustrating its proficiency in managing substantial alterations to the form and structure of deforming objects while preserving the integrity of the original camera and object movements.

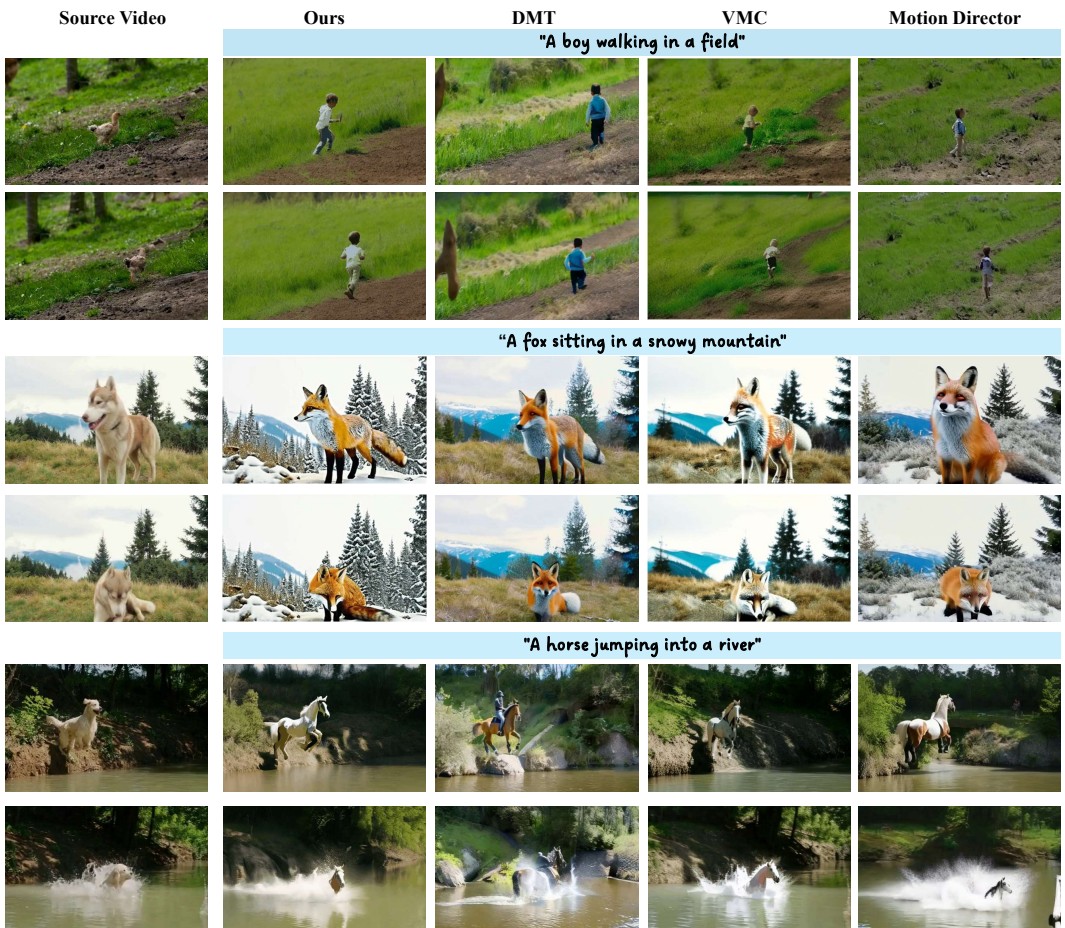

Figure 5: **Qualitative results**. Compared to DMT (Yatim et al., 2023), VMC (Jeong et al., 2023), and Motion Director (Zhao et al., 2023), our method not only preserves the original video's motion trajectory and object poses but also generates visual features that align with text descriptions.

### 4.1 QUALITATIVE EVALUATION

As illustrated in Figure 5, our method offers a qualitative enhancement over baseline approaches. It excels in preserving the motion trajectory and the object poses of the original video, as evidenced by the consistent positioning and posture of objects between the initial and final frames. Additionally, our technique demonstrates remarkable precision in generating visual features that are congruent with textual descriptions. For instance, in the scenario "a boy walking in a field", our model adeptly transforms a "walking duck" into a "walking boy", while preserving the original movement trajectory. In another instance, "a fox sitting in a snowy mountain", our approach adeptly embodies the essence of a snow-capped mountain scene with high motion fidelity. In contrast, while Motion Director (Zhao et al., 2023) is capable of producing similar visual features of the snowy mountain, it does not maintain the motion integrity of the original video as effectively as our method.

### 4.2 QUANTITATIVE EVALUATION

To thoroughly evaluate our method against baselines, we conducted assessments across multiple dimensions:

**Text Similarity.** Following the precedent set by previous research (Geyer et al., 2023; Esser et al., 2023; Jeong et al., 2023; Yatim et al., 2023), we utilize CLIP (Radford et al., 2021) to assess frame-

| Method | Text Similarity ↑ | Motion Fidelity ↑ | Temporal Consistency ↑ | FID ↓ | User Preference ↑ |
|---|---|---|---|---|---|
| DMT (Yatim et al., 2023) | 0.2883 | 0.7879 | 0.9357 | 614.21 | 16.19% |
| VMC (Jeong et al., 2023) | 0.2707 | 0.9372 | **0.9448** | 695.97 | 17.18% |
| MD (Zhao et al., 2023) | 0.3042 | 0.9391 | 0.9330 | 614.07 | 27.27% |
| **Ours** | **0.3113** | **0.9552** | 0.9354 | **550.38** | **39.35%** |

Table 1: **Quantitatve comparisons with existing methods.**

to-text similarity, calculating the average score to determine the accuracy of the edits in reflecting the intended textual modifications.

**Motion Fidelity.** To evaluate motion transfer effectiveness, we employ the Motion Fidelity Score introduced by (Yatim et al., 2023). This metric, which utilizes tracklets computed by an off-the-shelf tracking model (Karaev et al., 2023), measures the similarity between the motion trajectories in the unaligned videos, thus assessing how faithfully the output retains the input's motion dynamics. The Motion Fidelity Score is defined as:

$$\frac{1}{m} \sum_{\widetilde{\tau} \in \widetilde{\mathcal{T}}} \max_{\tau \in \mathcal{T}} \text{corr}(\tau, \widetilde{\tau}) + \frac{1}{n} \sum_{\tau \in \mathcal{T}} \max_{\widetilde{\tau} \in \widetilde{\mathcal{T}}} \text{corr}(\tau, \widetilde{\tau}), \quad (6)$$

where $\text{corr}(\tau, \widetilde{\tau})$ indicates the normalized cross-correlation between tracklets $\tau$ from the input and $\widetilde{\tau}$ from the output.

Those metrics above are considered for evaluating motion transfer tasks: conformance to the motion of the source video and the depiction of the appearance described by the text prompts. In addition to these primary metrics, quality evaluation is also conducted.

**Temporal Consistency.** Temporal consistency is widely used in video editing tasks to measure the smoothness and coherence of a video sequence (Jeong et al., 2023; Zhao et al., 2023; Kahatapitiya et al., 2024; Wu et al., 2023; Chen et al., 2023b). It is quantified by computing the average cosine similarity between the CLIP image features of all frame pairs within the output video.

**Fréchet Inception Distance (FID).** The Fréchet Inception Distance (FID), widely recognized for measuring the quality of images produced by generative models (Heusel et al., 2017), is applied in our study. In our case, images derived from a selection of 89 videos from the DAVIS dataset (Perazzi et al., 2016) are used as the reference set.

**User Study.** To rigorously evaluate our method's effectiveness, we designed a user study that involved **121** participants. They were presented with 10 randomly selected source videos paired with corresponding text prompts, creating **10** unique scenarios that test the versatility of our approach under varied conditions. For each scenario, participants were shown a set of 4 videos, each generated by a different method but under the same conditions of motion and text prompts. The survey specifically asked participants to identify the video that best conformed to the combination of the source video's motion and the textual description provided.

Table 1 presents a summary of the results for each metric. Evaluations were performed on a set of **66** video-edit text pairs, comprising **22** unique videos, for all metrics except user preferences. Both our method and Motion Director (Zhao et al., 2023) scored highly in text similarity. However, our approach surpassed Motion Director in motion fidelity, reinforcing the findings of the qualitative analysis. With respect to video quality, our method demonstrated a slight lag in temporal consistency when compared to VMC (Jeong et al., 2023), attributable to a lesser number of parameters. Nonetheless, in terms of individual frame quality, VMC was the least effective, significantly underperforming compared to our method. In the user study, our approach garnered a preference rate of 39.35%, the highest among the four methods evaluated, which further substantiates our method's proficiency in preserving the original video's motion and responding to text prompts.

### 4.3 ABLATION STUDY

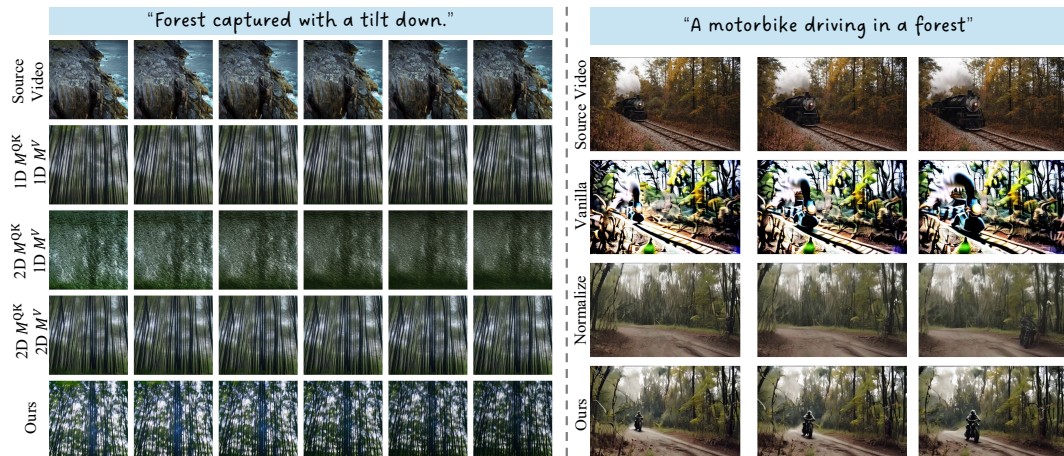

Figure 7: **Visual Result of the Ablation Study**. Left: Ablation of motion embedding design; Right: Ablation of inference strategy. **For better visualization, refer to the videos in the supplementary files**.

We conducted an ablation study of our method from two key perspectives: the design of motion embeddings and the inference strategy. For the **motion embedding design**, we evaluated three configurations: **(a)** spatial-1D $\mathbf{m}_i^{QK}$ with spatial-1D $\mathbf{m}_i^{V}$, **(b)** spatial-2D $\mathbf{m}_i^{QK}$ with spatial-1D $\mathbf{m}_i^{V}$, **(c)** ours, and **(d)** spatial-2D $\mathbf{m}_i^{QK}$ with spatial-2D $\mathbf{m}_i^{V}$. For the **inference strategy**, we compared our results with two approaches: **(e)** normalize, which reduces the mean value from $\mathbf{m}_i^{V}$, and **(f)** vanilla, which

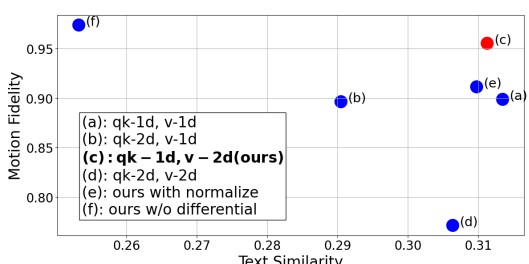

Figure 6: **Ablation Study.**

does not use the debias operation defined in Equation 5. The results are shown in Figure 6. The results demonstrate that our motion embedding design achieves a strong balance between capturing the motion of the original videos and generalizing well to diverse text prompts, reducing overfitting. Furthermore, after adopting our inference strategy, the text-to-video similarity is significantly improved.

## 4.4 LIMITATIONS

Our performance relies on the generative priors acquired by the T2V model. Consequently, the interplay between a specific target object and the motion in the input video may occasionally fall outside the T2V model's training distribution. On the other hand, our method may encounter challenges when the input video contains interfering motions from multiple objects, as this can affect the quality of our motion embedding. This is because we learn the overall motion from the entire video rather than focusing on the motion of a specific instance. Addressing this limitation by isolating instance-level motion is a potential area for future improvement.

## 5 CONCLUSION

In conclusion, we presented a novel approach to motion customization in video generation, addressing the challenge of motion representation in generative models. Our Motion Embeddings efficiently capture both global and spatial motion while preserving temporal coherence. Additionally, our inference strategy ensures motion-focused customization by removing appearance influences. Extensive experiments demonstrate the effectiveness of our method, providing a strong foundation for future advancements in instance-level motion learning.

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
