# OpenReview forum: "Motion Inversion for Video Customization"
_ICLR.cc/2025/Conference — Submitted to ICLR 2025_

### Official Review · Reviewer_qixD · 2024-10-20

**Soundness:** 2
**Presentation:** 3
**Contribution:** 1
**Rating:** 5
**Confidence:** 5

**Summary:**

This work introduces Motion Embeddings, a method for enhancing motion customization in video generation. The approach proposes a motion representation with two types of embeddings: Motion Query-Key Embedding for temporal attention modulation and Motion Value Embedding for adjusting attention values. An inference strategy is also proposed to focus exclusively on motion by excluding spatial dimensions and using a differential operation.

**Strengths:**

1. The proposed techniques for motion representation are clear and reasonable.

2. The experiments are comprehensive, featuring a variety of qualitative and quantitative results.

3. The paper is well-written and easy to understand.

**Weaknesses:**

1. The primary concern is that the proposed method's effectiveness has already been demonstrated in previous works, such as MotionDirector [1]. For instance, MotionDirector employs motion LoRA in the temporal layer for motion representation, which essentially involves adding weights to the Q, K, and V projectors (Wq, Wk, and Wv in Eq. 3). This paper introduces learnable embeddings added to the input features in Eq. 3. While these two embedding approaches are not identical, their functionality appears similar, thus limiting the novelty of the contribution.

2. The interpretability of the learned embeddings is insufficiently addressed. Although the paper claims improved interpretability, no convincing evidence is provided. For instance, could the authors visualize the learned QK embeddings to determine if the attention maps exhibit meaningful patterns? Alternatively, are the learned V embeddings interpretable in any discernible way?

[1] MotionDirector: Motion Customization of Text-to-Video Diffusion Models. Zhao et al., 2024.

**Questions:**

1. The paper claims to utilize 1-D QK embeddings for global motion and 2-D V embeddings for spatially varying movements. Could the authors provide results demonstrating their effectiveness when these embeddings function independently?

2. The proposed method targets temporal attention, while the base models decompose attention into separate spatial and temporal components. It would be more compelling to demonstrate the effectiveness of this technique within full attention architectures, where spatial and temporal attention are integrated into a single attention mechanism (e.g., Open-Sora [1]).

[1] Open-Sora: Democratizing Efficient Video Production for All, Zheng et al., 2024.

**Details Of Ethics Concerns:**

No ethics concerns.

---

> ### Author Response · Authors · 2024-11-20
>
> Thank you for your thoughtful and detailed feedback. We sincerely appreciate the opportunity to clarify and emphasize the contributions of our work and address the concerns raised.
>
> ### **Weakness 1**
> We sincerely thank the reviewer for raising concerns about the potential similarities between our proposed method and MotionDirector [1]. While both approaches utilize embeddings for motion representation, we would like to highlight several key differences in design and functionality that distinguish our method and address critical limitations in existing techniques:
>
> 1. **Core Contributions**:  Our method introduces a novel combination of **1D embeddings for QK** and **2D embeddings for V**, along with a **differential operation** to effectively debias static appearance. Through ablation studies, we show that vanilla motion embeddings alone perform suboptimally, whereas our carefully designed embeddings lead to substantial performance improvements.
> 2. **Robustness to Optimization Targets**: Our method demonstrates greater robustness compared to MotionDirector when subjected to different optimization targets. As shown in the table below, implementing MotionDirector with a vanilla loss results in a more significant drop in performance, while our method experiences only minor degradation. This highlights the stability and reliability of our approach.
>
> | Method| Metric|Hybrid Loss|Vanilla MSE Loss|
> |--|--|--|--|
> | **MotionDirector** | Text Similarity| 0.304| 0.295 |
> || Motion Fidelity|0.939| 0.930|
> | **Ours**| Text Similarity| 0.311| 0.310|
> ||Motion Fidelity|0.955|0.951|
>
> 3. **Compatibility with Full Attention DiT Models**: MotionDirector relies on specialized loss functions that require the model architecture to use a mixture of 1D temporal attention and 2D spatial attention. In contrast, our method does not depend on such architectural constraints, making it more flexible. Modern, full-attention DiT video models integrate spatial and temporal attention into a unified mechanism, which renders MotionDirector’s loss functions incompatible. However, our approach remains fully compatible with these state-of-the-art models due to its adaptability to general attention mechanisms.
>
> We hope these points clarify the unique contributions of our work and address the concerns raised. We believe that our method offers a meaningful advancement in the field of motion representation by providing a robust and versatile framework.
>
> ### **Weakness 2**
> We appreciate the reviewer’s concern regarding the interpretability of the learned embeddings and we strive to provide insights based on our experiments.
>
> Visualizing the learned QK and V embeddings is inherently challenging due to the latent dimensionality, which makes them less straightforward to interpret compared to traditional attention maps. Nonetheless, we demonstrate the effectiveness of these embeddings through the following experiments. Corresponding videos are included in the supplementary materials  (The *Embedding Ablation* section in newly updated supp/index.html):
>
> 1. **1D QK Embeddings**:
>    Our empirical results show that 1D QK embeddings are effective at capturing global motion patterns. For instance, in a source video involving lateral panning, the 1D QK embeddings successfully capture the overall global movement, even in the absence of 2D V embeddings.
>
> 2. **2D V Embeddings**:
>    The 2D V embeddings, on the other hand, focus on capturing localized motion. For example, when only 1D QK embeddings are used, the model struggles to capture the movement of objects within the scene. However, with the inclusion of 2D V embeddings, the model accurately learns the motion dynamics of individual objects in the source video.
>
> We hope these experiments provide clarity on the roles and interpretability of the different embeddings.
>
> ### **Questions - Full Attention Architecture**
> Thank you for your valuable comment. We appreciate the suggestion to demonstrate the applicability of our method in full attention architectures.
>
> In full attention architectures, motion value embeddings inherently exhibit spatial-temporal properties. To adapt our method, we propose to retain the motion query and motion key embeddings as **1D embeddings**, which offer conceptual simplicity and computational efficiency. During the forward pass, these 1D embeddings can be expanded to spatial-temporal dimensions to align with the full attention mechanism.
>
> This design ensures that our approach remains compatible and effective, even in architectures that do not explicitly separate spatial and temporal attention. We believe this adaptability further demonstrates the flexibility and generalizability of our method.
>
> ---
>
> Thank you once again for your valuable feedback. We hope this response clarifies your concerns and highlights the unique contributions and flexibility of our work. We look forward to further discussions or clarifications if needed.

---

> > ### Comment · Reviewer_qixD · 2024-11-24
> >
> > Thank you for the answers.
> >
> > The visualization of the QK and V embeddings does enhance interpretability by clarifying their respective roles. However, it also raises some concerns. For instance, in the "A tiger walking in the forest" example, the debiasing capability of the V embedding appears ineffective. Specifically, it seems to copy textures directly from the source video rather than aligning them well with the prompts.
> >
> > My primary concern, however, remains with Weakness 1. The improvement achieved by this method compared to MotionDirector is relatively minor. Furthermore, the performance drop when switching from Hybrid loss to vanilla loss is not particularly significant and might even fall within the range of implementation variance. Most importantly, these two methods appear conceptually very similar, mainly in the addition of weights to the original temporal attention mechanism.

---

> > > ### Author Response · Authors · 2024-11-24
> > >
> > > We are very happy to address any of your concerns. Below is our updated response:
> > >
> > > ---
> > >
> > > ### Clarification: "Debiasing capability of the V embedding appears ineffective"
> > >
> > > The example provided, such as "A tiger walking in the forest," illustrates that the 2D V embedding effectively learns **local movements**, with the primary focus being **motion alignment**, not texture alignment. Additionally, we provide another example using the same motion embeddings to address concerns regarding **text alignment**. Quantitative evaluations of text alignment in our paper further strengthen our claims, providing more convincing evidence than a single example.
> > >
> > > ---
> > >
> > > ### Concern: Improvement achieved is minor
> > >
> > > 1. **New VBench Metrics**: On VBench metrics (Subject Consistency, Background Consistency, Motion Smoothness), **our method outperforms MotionDirector across all dimensions**. Notably, **motion fidelity** and **text similarity** improve by nearly 1%, which is significant for tasks requiring high precision.
> > >
> > > 2. **Controlled Comparisons**: To ensure fairness, we used the **same base model (ZeroScope)**, identical **hyperparameters** (CFG = 10, no negative prompt, 30 inference steps), **same prompts**, and **same seeds (0-10)**, eliminating implementation variance.
> > >
> > >    |  | **Text Similarity** | **Motion Fidelity** | **Temporal Consistency** | **Subject Consistency** | **Background Consistency** | **Motion Smoothness** | **FID** | **User Preference** |
> > >    | --- | --- | --- | --- | --- | --- | --- | --- | --- |
> > >    | **MotionDirector** | 0.3042 | 0.9391 | 0.9330 | 0.9127 | 0.9421 | 0.9689 | 614 | 27.27% |
> > >    | **Ours** | **0.3113** | **0.9552** | **0.9354** | **0.9236** | **0.9425** | **0.9696** | **550** | **39.35%** |
> > >
> > > The reviewer suggests these improvements are small and dismissible; however, outperforming across **all metrics**, including consistent improvements in motion fidelity and text similarity, highlights the robustness of our approach and should not be overlooked.
> > >
> > > ---
> > >
> > > ### Concern: Performance drop when switching from hybrid loss to vanilla loss is minor
> > >
> > > The experiments comparing hybrid loss and vanilla loss were conducted under identical conditions, with the **loss function as the only variable**. The findings are as follows:
> > >
> > > 1. **Observed Results**: Both **text similarity** and **motion fidelity** dropped by **1%**, a measurable yet minor loss. Despite this, our method demonstrates **robustness**, with almost no degradation.
> > > 2. **Flexibility Advantage**: Hybrid loss, MotionDirector’s core contribution, cannot be applied to full attention models, whereas our **optimization target** is flexible and broadly applicable.
> > >
> > > ---
> > >
> > > ### Concern: Conceptual similarity between methods
> > >
> > > We respectfully disagree that both methods are conceptually similar due to **"adding weights to the original temporal attention mechanism."** It is neither fair nor reasonable to reduce two works to a shared concept while ignoring their unique contributions. By this logic:
> > >
> > > 1. **MotionDirector vs. LoRA**: MotionDirector could be dismissed as conceptually similar to LoRA, ignoring its innovation in **hybrid loss** for appearance and motion debiasing.
> > > 2. **ControlNet vs. LoRA**: ControlNet could similarly be dismissed, overlooking its unique conditioning capabilities that extend Stable Diffusion's functionality.
> > > 3. **AnimateDiff vs. VideoCrafter**: These methods share temporal attention and video generation goals, yet AnimateDiff extends text-to-image generation to videos, while VideoCrafter builds a model trained from scratch.
> > >
> > > Such logic risks dismissing many **outstanding works** that share concepts but contribute unique advancements. Similarly, while our method and MotionDirector both modify temporal attention mechanisms, their contributions are fundamentally different:
> > >
> > > - **MotionDirector**: Innovates with its **hybrid loss** for appearance and motion debiasing.
> > > - **Our method**: Introduces a novel **optimization target** based on motion embeddings, validated through quantitative metrics and qualitative examples.
> > >
> > > Our **optimization target** and MotionDirector’s **hybrid loss** are **orthogonal contributions**. Our target can be combined with their hybrid loss, and we have shown it outperforms LoRA under the same optimization loss. Reducing our work to a shared concept oversimplifies its contributions and impact.
> > >
> > > ---
> > >
> > > If you have further doubts about any of the above points, we would be happy to continue the discussion.

---

> > > > ### Comment · Reviewer_qixD · 2024-11-24
> > > >
> > > > Thank you for your response.
> > > >
> > > > While I appreciate your explanation, I respectfully disagree with the logic regarding the conceptual similarity between methods. Methods like MotionDirector or ControlNet are conceptually innovative because they introduce new paradigms for addressing specific tasks. In contrast, LoRA primarily represents an implementation strategy. That said, the method under discussion shares a similar approach to solving a task with MotionDirector.
> > > >
> > > > To clarify, a direct comparison of the formulations for this method and MotionDirector is as follows:
> > > >
> > > > this method
> > > > \begin{equation}
> > > >     \text{TA}_i(\mathbf{F}) = \text{softmax}\left(
> > > >     \frac{\mathbf{W}_q(\mathbf{F} + \mathbf{m}_i^{QK})\mathbf{W}_k(\mathbf{F} + \mathbf{m}_i^{QK})^T}{\sqrt{d_k}}
> > > >     \right)\mathbf{W}_v(\mathbf{F} + \mathbf{m}_i^V),
> > > > \end{equation}
> > > >
> > > > MotionDirector:
> > > > \begin{equation}
> > > >     \text{TA}_i(\mathbf{F}) = \text{softmax}\left(
> > > >     \frac{(\mathbf{W}_q+\mathbf{W}_i^{Q})\mathbf{F}(\mathbf{W}_k+\mathbf{W}_i^{K})\mathbf{F}^T}{\sqrt{d_k}}
> > > >     \right)(\mathbf{W}_v+\mathbf{W}_i^V)\mathbf{F},
> > > > \end{equation}
> > > >
> > > > In terms of the learning process, the optimization of weights in this method also appears to follow a widely adopted approach, closely aligned with the optimization strategy used in LoRA tuning—please correct me if I am mistaken.
> > > >
> > > > To this end, my primary concern remains unresolved.

---

> > > > > ### Author Response · Authors · 2024-11-24
> > > > >
> > > > > Thank you for your response. We appreciate your feedback and would like to further discuss it:
> > > > > ### Correcting LoRA Formulations
> > > > >
> > > > > The reviewer's understanding of LoRA seems to be incorrect. Specifically, in the context of $qkv$ projections, the linear layers cannot be directly added to additional parameters. The correct formulation for LoRA adjustments should be:
> > > > >
> > > > > \begin{equation}
> > > > > \text{TA}_i(\mathbf{F}) = \text{softmax}\left(
> > > > > \frac{\left[(\mathbf{W}_q \mathbf{F}) + \mathbf{A}_q (\mathbf{B}_q \mathbf{F})\right]
> > > > > \left[(\mathbf{W}_k \mathbf{F}) + \mathbf{A}_k (\mathbf{B}_k \mathbf{F})\right]^T}{\sqrt{d_k}}
> > > > > \right)
> > > > > \left[(\mathbf{W}_v \mathbf{F}) + \mathbf{A}_v (\mathbf{B}_v \mathbf{F})\right].
> > > > > \end{equation}
> > > > >
> > > > > Here:
> > > > > - $\mathbf{A}_q$, $\mathbf{A}_k$, and $\mathbf{A}_v$ are learnable projection matrices that project from a lower-dimensional space back to the original high-dimensional space.
> > > > > - $\mathbf{B}_q$, $\mathbf{B}_k$, and $\mathbf{B}_v$ are learnable projection matrices that project the input $\mathbf{F}$ into a lower-dimensional space.
> > > > >
> > > > > This formulation explicitly demonstrates how LoRA operates through additional parameterization without directly summing weights.
> > > > >
> > > > > ---
> > > > >
> > > > > ### Conceptual Similarity Between Methods
> > > > >
> > > > > The reviewer argues that **MotionDirector** and our work share similarities because temporal attention is affected by learning objectives. This argument dismisses our previous examples, and we respond with a more concrete comparison:
> > > > >
> > > > > In our paper, we discuss **VMC (Video Motion Customization)**[1], which addresses the same task—**Motion Customization**—but uses a distinct approach. VMC employs a **motion vector loss** rather than MotionDirector's hybrid loss and directly optimizes the parameters of $\mathbf{W}_q$, $\mathbf{W}_k$, and $\mathbf{W}_v$. The formulation for VMC's optimization can be expressed as:
> > > > >
> > > > > \begin{equation}
> > > > > \text{TA}_i(\mathbf{F}) = \text{softmax}\left(
> > > > > \frac{\left[(\mathbf{W}_q + \Delta\mathbf{W}_q)\mathbf{F}\right]\left[(\mathbf{W}_k + \Delta\mathbf{W}_k)\mathbf{F}\right]^T}{\sqrt{d_k}}
> > > > > \right)\left[(\mathbf{W}_v + \Delta\mathbf{W}_v)\mathbf{F}\right].
> > > > > \end{equation}
> > > > >
> > > > > Here:
> > > > > - $\Delta\mathbf{W}_q$, $\Delta\mathbf{W}_k$, and $\Delta\mathbf{W}_v$ represent the parameters directly optimized using motion vector loss.
> > > > >
> > > > > Based on the reviewer's logic, if we ignore the differences in loss design, **VMC** and **MotionDirector's optimization of weights** share similarities, **which would suggest that one should be dismissed**.
> > > > >
> > > > > ---
> > > > >
> > > > > ### We Wish the Reviewer to Rethink
> > > > >
> > > > > 1. **Differences in Contributions**:
> > > > >    - Our work is **orthogonal** to both VMC and MotionDirector.
> > > > >    - Our **motion embedding** can integrate the core contributions from both VMC and MotionDirector.
> > > > >
> > > > > 2. **Performance Improvements**:
> > > > >    - Our approach demonstrates significant performance gains, which cannot be dismissed by focusing only on similarities in optimization methods.
> > > > >
> > > > > 3. **Insights Provided by Our Work**:
> > > > >    - Our method introduces valuable insights into the role of motion embedding in temporal attention, which has not been explored in prior work.
> > > > >
> > > > > ---
> > > > >
> > > > > ### Conclusion
> > > > >
> > > > > We emphasize that sharing some commonalities with prior works does not diminish the novelty of our contributions. Each approach—whether VMC, MotionDirector, or our method—has unique design choices and insights, and these differences should not be overlooked.
> > > > >
> > > > > **Reference**:
> > > > > [1] VMC: Video Motion Customization using Temporal Attention Adaption for Text-to-Video Diffusion Models

---

> > > > > > ### Comment · Reviewer_qixD · 2024-11-25
> > > > > >
> > > > > > Thank you for the response.
> > > > > >
> > > > > > Your explanation about LoRA seems unnecessary. After merging, your formulation of LoRA closely resembles mine - they are all linear operations.
> > > > > >
> > > > > > VMC and MotionDirector are concurrent works that were released last year. However, your work cannot be classified as concurrent with them.
> > > > > >
> > > > > > I believe that if the proposed embedding could be extended to other, more intriguing task domains because of its distinct interpretability (maybe still not enough in this work), its contributions would gain broader recognition.

---

> ### Author Response · Authors · 2024-11-25
>
> Thank you for your detailed feedback. We appreciate your insights and would like to clarify a few points further.
>
> ---
>
> ### The necessity of explaining LoRA formulations ###
>    We believe **it is essential to explicitly write out the LoRA formulations to highlight the differences between MotionDirector and VMC**. Without this clarity, these works might appear indistinguishable in terms of their formulations, which would fail to acknowledge their unique contributions.
>
> ---
>
> ### The definition of "concurrent work" ###
>    We are curious about the criteria for defining "concurrent" work. If the definition is based on the publication dates at major conferences, **MotionDirector and VMC  were published in different years**. Alternatively, if it is based on the submission dates to platforms like arXiv, numerous recent works on motion customization share similarities in formulations. By this logic, many of these works could be dismissed under the same reasoning.
>
>    However, we emphasize that merely adopting a specific formulation does not inherently guarantee effectiveness. As demonstrated through both qualitative and quantitative evaluations in our work, the superior performance of our motion customization approach is a direct result of our unique design, which advances the state-of-the-art (SOTA) in this domain.
>
> ---
>
> ### The evolution of concerns ###
>    We also observe an evolution in the reviewer's concerns:
>    - **Initial Stage:** The primary concern was conceptual similarity with MotionDirector and a perceived lack of interpretability.
>    - **Second Stage:** After our first rebuttal, the reviewer stated, "My primary concern remains with weakness 1," dismissing our improvements across comprehensive metrics and attributing them to implementation variance. The discussion shifted to conceptual similarity, with the reviewer suggesting that since MotionDirector and VMC are concurrent works, their similarity is acceptable. However, this appears to rely on the reviewer's subjective definition of "concurrent," which remains unclear despite our efforts to clarify.
>    - **Third Stage:** The primary concern now appears to have reverted to interpretability, despite the reviewer initially acknowledging that "The visualization of the QK and V embeddings does enhance interpretability by clarifying their respective roles."
>
> ---
> ### Conclusion ###
> The inconsistent shifts in the reviewer’s concerns, from similarity to variance and back to interpretability—despite initial acknowledgment of our interpretability improvements—make the feedback confusing and unclear.

---

> > ### Comment · Reviewer_qixD · 2024-11-26
> >
> > Thanks for the answers.
> >
> > My main concern never shifts, which at the beginning is the conceptual similarity with previous works, i.e., MotionDirector.
> >
> > Regarding the LoRA formulation, while it can be expressed in various ways, achieving a better understanding of its actual functionality requires a comparison in a more uniform framework. Such an approach would help clarify the unique aspects of the method.
> >
> > On interpretability, we acknowledge that the proposed method enhances it; however, the improvement may not yet elevate it to a level that introduces a more intriguing feature beyond the motion-following task. My previous comment on interpretability was merely a suggestion—if the authors can explore and identify a distinct interpretability aspect, the concern regarding similarity becomes less significant, as the work would then contribute in a uniquely valuable way.
> >
> > Regarding the concept of concurrent work, it is true that no clear definition exists. However, the idea of tuning temporal weights for motion mimicry was demonstrated to be effective and widely acknowledged over a year ago. Unfortunately, this paper does not offer significantly new insights beyond that established foundation.
> >
> > So I will keep my rate.

---

> ### Author Response · Authors · 2024-11-27
>
> Thank you for your feedback. We appreciate that we’ve reached a common understanding with the reviewer regarding the fact that our method may not offer **significantly** new insights beyond the established foundation. However, we believe that our work provides valuable insights that are worth sharing with the broader research community. These findings can serve as a stepping stone for future work. We would like to reiterate the distinction between our motion embeddings and the temporal LoRA weights, which might help clarify the unique aspects of our approach:
>
> 1. **Parameters vs. Embeddings**: As we addressed in our response to Reviewer 5Vdk, while temporal LoRA does not explicitly model a temporal dimension, our motion embeddings are specifically designed to capture such temporal features. This design makes the learning of temporal features more effective, as evidenced by both our quantitative and qualitative results. Previous works [1, 2] have shown that although optimizing parameters and optimizing latent representations can complete same task, optimizing latent representations avoids the indirect and potentially entangled effects that parameter-level adjustments can introduce.
>
> 2. **Positional Embeddings**: Additionally, our motion embeddings function more like learnable positional embeddings rather than just additional features. This was actually one of the original motivations behind our approach. We observed that directly extrapolating absolute positional encodings in video models allowed us to control motion intensity, successfully extending the generated video (see **supp/index.html/Effect of Positional Embedding of Video Generation Models**). This led us to refine our approach, ultimately designing the motion embeddings to enable more precise control over motion, and we believe this represents a meaningful step forward in motion modeling.
>
> Finally, we fully respect your evaluation, which sets a very high standard. At the same time, we value our contributions and are confident that these findings will be beneficial to the research community, providing insights that could pave the way for more significant advancements in the future.
>
> References
> [1] StyleClip: Text-Driven Manipulation of StyleGAN Imagery
>
> [2] StyleGAN-NADA: CLIP-Guided Domain Adaptation of Image Generators

---

### Official Review · Reviewer_5Vdk · 2024-10-21

**Soundness:** 3
**Presentation:** 3
**Contribution:** 3
**Rating:** 5
**Confidence:** 4

**Summary:**

This paper introduces a novel method for motion customization in video generation, addressing the underexplored challenge of motion representation in video generative models. The proposed approach leverages Motion Embeddings—temporally coherent representations derived from video data—to enhance the motion modeling in video diffusion models without compromising spatial integrity. The paper presents two key components: the Motion Query-Key Embedding, which modulates the temporal attention map, and the Motion Value Embedding, which adjusts the attention values. Additionally, the paper proposes an inference strategy that removes spatial dependencies and focuses the embeddings purely on motion. Through extensive experiments, the method demonstrates its ability to effectively customize motion within videos while maintaining efficiency and spatial consistency.

**Strengths:**

1. Superiority against sota baselines, where evaluation is done both quantitatively and qualitatively.
2. Novel task
3. Clarity of the presentation and writing
3. Novelty in introducing the motion embedding concept
4. Easy to follow the paper

**Weaknesses:**

1. I understand the main method composed of two part; (1)What they optimize and (2)How they optimize.
In terms of the optimization target (1), there exists technical contribution. But in terms of the loss calculation (2), they either use the standard denoising loss, or the VMC/MotionDirector [1,2] loss.
If my understanding above is right, I think the paper should claim more on optimizing the motion embeddings are so important and superior than other options (temporal attention / temporal lora).

2. Finetuning temporal attn would directly hurt the model's original prior but training temporal lora would avoid this problem. Thus, the paper should show theoretical understanding / empirical insights that their eq(3) is superior to temporal lora optimization.

3. L260 ~ L264 need more explanation. What is is trying to achieve from the modulation? If the goal is to achieve appearance debiasing, why would it work? Is this modulation ablated in the paper?

4. In Figure 4 and in the paper in general, what does "excluding spatial dimensions" mean? Do the authors refer to "optimizing temporal attn related parameters instead of self attn related parameters? In Figure 4 left, what are the authors visualizing in the middle? Attention maps of the same module in the t2v unet? And how did they visualize them (e.g. PCA)?

5. In Fig 4 right, what do the authors mean by "differential"? If they mean residuals across frame-axis? If so, using different phrase would be more appropriate.

6. In quantitative evaluation, for computing temporal consistency, clip is not the optimal choice. Referencing metrics from V-bench [3] would increase the value of the paper.

[1] VMC: Video Motion Customization using Temporal Attention Adaption for Text-to-Video Diffusion Models, CVPR 2024

[2] MotionDirector: Motion Customization of Text-to-Video Diffusion Models, ECCV 2024

[3] VBench: Comprehensive Benchmark Suite for Video Generative Models, CVPR 2024

**Questions:**

Please address the Weakness section

---

> ### Author Response · Authors · 2024-11-20
>
> Thank you for your detailed feedback. We appreciate the opportunity to clarify the contributions of our work.
>
> ## Weakness 1 & 2
> ### **1. Theoretical Superiority of Our Optimization Target**
> The learning of motion can be interpreted as an overfitting process to a single video while simultaneously removing appearance information (i.e., disentangling motion from appearance).
> - We introduce **1D embeddings for QK** and **2D embeddings for V**, paired with a **differential operation** that effectively debiases static appearance. The **1D QK embedding** helps avoid overfitting to the edges and contours of the source video, while the **differential operation** for the 2D value embedding removes static appearance, focusing purely on motion dynamics.
> - Unlike methods such as MotionDirector, which rely heavily on the architecture of the video model, our method makes no assumptions about the video generation model's architecture, making it more generalizable.
>
> ### **2. Empirical Superiority of Our Optimization Target**
> Our approach allows the learned motion representation to be effectively combined with various text prompts, leading to high text similarity. We provide evidence for this through the ablation study and results presented in the main paper:
> - As shown in **Table 1 in paper**, when using the **same loss** but different optimization targets (e.g., LoRA, QKV projection with VMC, alternative motion embeddings), our method consistently achieves superior results, with high text similarity and high motion fidelity.
> - Moreover, when employing the **same optimization target** but with different loss functions—particularly in comparison with MotionDirector's temporal LoRA—the performance of other methods degrades more than ours. This highlights the robustness and effectiveness of our optimization strategy.
>
> | Method|Metric| Hybrid Loss | Vanilla MSE Loss|
> |--|--|--|--|
> | **motion director** | Text Similarity| 0.304| 0.295|
> || Motion Fidelity| 0.939| 0.930|
> | **ours**| Text Similarity| 0.311| 0.310|
> || Motion Fidelity| 0.955| 0.951|
>
> These points collectively reinforce the importance of optimizing motion embeddings over approaches like temporal attention or temporal LoRA, demonstrating both theoretical soundness and empirical superiority in preserving the model's original prior and achieving higher-quality motion representation.
>
> ## Weakness 3
> The operation debiases static appearance by leveraging principles similar to optical flow, where differences between consecutive frame embeddings highlight motion while canceling static features. Static appearance information, such as background textures, tends to remain constant across frames, and the differencing effectively removes these redundancies. This ensures that the embeddings focus solely on temporal dynamics, discarding irrelevant appearance information. By isolating motion, the operation improves generalization, prevents static biases from dominating, and enhances motion fidelity. Our ablation studies validate that this significantly improves motion representation quality compared to before.
> ## Weakness 4
> ### **1. Visualization of Attention Maps in Figure 4 (Left)**
> In **Figure 4 (left)**, we visualize attention maps generated using different types of QK motion embeddings:
> - **1D and 2D QK Embeddings**:
>     - The **2D QK embedding** is structured as \((H \times W, N, C)\), where \(H\) and \(W\) represent spatial dimensions, \(N\) represents the number of frames, and \(C\) represents the feature channels.
>     - For the visualization, we extracted attention maps of shape \((N \times N, H \times W)\). This allowed us to visualize \(N \times N\) spatial maps of size \(H \times W\). These maps revealed that the 2D QK embedding tends to capture all edge information across video frames. However, this detailed spatial information could affect motion customization, especially when generalizing to significantly different shapes (e.g., transforming a tank into a bicycle).
>     - To address this limitation, we excluded the spatial dimension and designed a **1D QK embedding**, which avoids overly specific spatial information while retaining essential motion dynamics.
>
> ### **2. Visualization Methodology**
> - The attention maps of shape \((N \times N, H \times W)\) were aggregated across:
>     - All diffusion steps,
>     - Attention heads, and
>     - Layers within the model.
> - The aggregated attention maps were then treated as an image for visualization purposes.
>
> ## Weakness 5
> Thank you for your suggestion. We have revised this section in the updated manuscript.
>
> ## Weakness 6
> We include the evaluation using temporal quality metrics from **vbench** within multiple dimensions:
>
> | Method| Subject Consistency | Background Consistency | Motion Smoothness |
> |--|--|--|--|
> | **DMT**| 0.9169| 0.9359| **0.9711**|
> | **VMC**| 0.9209| 0.9226| 0.9683|
> | **MotionDirector**| 0.9127| 0.9421| 0.9689|
> | **Ours**| **0.9236**| **0.9425**| 0.9696|

---

> > ### Author Response · Authors · 2024-11-24
> >
> > Dear Reviewer,
> >
> > Thank you again for your valuable feedback and thoughtful comments during the discussion phase. We would like to kindly remind you that the discussion period will conclude on **November 26th**. If you have any additional questions, concerns, or clarifications you would like us to address, we would be more than happy to provide prompt responses.
> >
> > Your insights have been instrumental in shaping the final version of our submission, and we greatly appreciate your time and effort in engaging with our work.
> >
> > Thank you for your attention, and we look forward to hearing from you!

---

> > > ### Comment · Area_Chair_av9N · 2024-11-24
> > > **Discussion Period Ending Soon**
> > >
> > > Dear Reviewer 5Vdk,
> > >
> > > The discussion period will end soon. Please take a look at the author's comments and begin a discussion.
> > >
> > > Thanks,
> > > Your AC

---

> > > > ### Comment · Reviewer_5Vdk · 2024-11-25
> > > > **RE: Discussion Period Ending Soon**
> > > >
> > > > Dear AC,
> > > >
> > > > I will carefully re-read all the discussion and the revisions then present my discussion.
> > > >
> > > > Thank you.

---

> > ### Comment · Reviewer_5Vdk · 2024-11-25
> > **Response by Reviewer**
> >
> > Thank you for your addressing my concerns and for conducting additional experiments.
> > In overall, the qualitative results are impressive and quantitatively superior than DMT, VMC, MotionDirector. However, similar to Reviewer qixD, my concern regarding the optimization target persists. Why should optimizing Motion Inversion's motion embeddings (additional features for temporal attention features) be better than Motion Director's temporal lora weights (additional weights for temporal attention weights)?
> >
> > I have a few additional questions:
> > 1. Can the authors compare time/memory consumption between Motion Inversion, MotionDirector, VMC, DMT?
> > 2. Is L in eq. (2) defined in advance?
> > 3. For the DDIM Inversion part, how many steps are used? If everything else is fixed but the steps are decreased, how does Motion Inversion perform?
> > 4. For the authors' response above (2. Empirical Superiority of Our Optimization Target), especially for the "using the same loss but different optimization targets" case, can the authors provide visuals? As pointed by Reviewer qixD, the improvements are marginal.
> > 5. This is a minor point but again, regarding phrasing practices, I believe using "differential" for what the authors are trying to deliver is inappropriate.

---

> > > ### Author Response · Authors · 2024-11-26
> > >
> > > Thank you for your feedback. We have updated the manuscript, and our responses are provided below:
> > >
> > > ---
> > >
> > > ### Why motion embeddings better than temporal lora?
> > >
> > > Temporal LoRA weights inherently **lack an explicit temporal dimension**, which limits their ability to capture temporal-specific characteristics effectively.
> > > In contrast, our motion embedding is inherently temporal-aware, incorporating a dedicated temporal dimension that captures variance along the temporal axis.
> > > This enables our approach to better model and learn the temporal distribution patterns of the source video.
> > >
> > > For example, a 1D embedding, such as a sequence of scalar values, cannot effectively encode spatial features like contours or shapes when compared to a 2D embedding, which captures the relationships between rows and columns in an image.
> > > Similarly, temporal LoRA weights, which do not have an explicit temporal structure, are less capable of encoding temporal dependencies and dynamics compared to our motion embedding, which is specifically designed to address these aspects.
> > >
> > > Also, we also improve the design of temporal-aware embeddings.
> > > Our unique design, in combination with our debiasing strategy, that makes these "additional features" a superior choice for motion representation.
> > >
> > >
> > >
> > > ---
> > >
> > > ### Additional Question 1 - Time/Memory Consumption
> > > We provide a detailed comparison of time and memory consumption between our method (Motion Inversion) and other methods (MotionDirector, VMC, DMT) in the table below.
> > > | **Method**         | **Train Time (Minutes) ↓** | **Train Mem (MB) ↓** | **Test Time (Seconds) ↓** | **Test Mem (MB) ↓** |
> > > |---------------------|----------------------------|-----------------------|---------------------------|---------------------|
> > > | Base Model          | -                        | -                     | 33.60                    | 6228                |
> > > | DMT (CVPR24)        | 7.31                     | **6228**              | 268.98                   | 12945                   |
> > > | VMC (CVPR24)        | 8.65                     | 32263                 | **33.60**                | **6228**            |
> > > | Motion Director     | 6.83                     | 16153                 | 37.81                    | 8189                |
> > > | **Ours**            | **5.71**                 | 13367                 | 33.71                    | 6388                |
> > >
> > >
> > >
> > > ---
> > >
> > > ### Additional Question 2 - Definition of L
> > > The definition of L can be found in L242–243 of the paper.
> > >
> > > ---
> > > ### Additional Question 3 - DDIM Inversion steps
> > > The default number of steps is 50.
> > > Please refer to the additional results provided(**supp/index.html/Effect of Different Numbers of DDIM Inversion Steps**).
> > > Note that the number of DDIM inversion steps has a subtle influence on the final outcome.
> > >
> > >
> > > ---
> > > ### Additional Question 4 - Visual Comparison
> > > Please refer to **supp/index.html/Comparison of Loss and Optimization Target** for visual comparisons.
> > > After applying different loss, methods like MotionDirector exhibit noticeable issues in text responsiveness—for example, failing to properly generate "burning" effects.
> > > Additionally, motion fidelity is affected, as seen in the "elephant" example, where the walking motion is not accurately replicated from the source video.
> > > In contrast, our method maintains consistent performance across both text responsiveness and motion fidelity, demonstrating its robustness.
> > >
> > >
> > > ---
> > > ### Additional Question 5 - "Differential"
> > > Thank you for your feedback.
> > > We appreciate your suggestion and are actively seeking a more suitable term to replace "differential."
> > > For now, we plan to use "debias" as a temporary alternative unless a better term is identified.

---

> > > > ### Comment · Reviewer_5Vdk · 2024-11-26
> > > > **Response by Reviewer**
> > > >
> > > > Thank you for addressing my concerns and for the detailed experiments. Accordingly, I raised my score to 5.
> > > > I do not agree that temporal LoRA weights are inherently lacking temporal dimension. I believe they do inherently model temporal dimension just like temporal self attention does, but the rank is lower than the full attention weights. Please correct me if I am wrong.

---

> > > > > ### Author Response · Authors · 2024-11-26
> > > > >
> > > > > Thank you for your thoughtful comments and for engaging in a detailed discussion about the temporal modeling capabilities of LoRA parameters. We sincerely appreciate your insights.
> > > > >
> > > > > In our earlier response, we stated that Temporal LoRA “inherently lacks temporal dimensions.” Upon reflection, we recognize that this wording may have been misleading. Temporal LoRA does indeed model temporal dimensions by learning additional weights that adjust temporal attention mechanisms. However, it does so in an **indirect** manner, as these learned weights influence the parameters of temporal attention rather than directly encoding temporal features. In contrast, our method introduces motion embeddings that explicitly and **directly** encode temporal features. This direct modeling allows for a more disentangled representation of spatiotemporal dynamics compared to the parameter-level adjustments in Temporal LoRA, which may be more entangled with the original attention weights.
> > > > >
> > > > > To further illustrate this distinction, we draw an analogy with findings in the StyleGAN literatures (while these studies use earlier-generation models, we believe the principles are broadly applicable). Prior works [1, 2] demonstrate that semantic control can be achieved by either optimizing network parameters [2] or by directly optimizing latent representations [1]. Editing in the latent space often provides more disentangled and interpretable results, as latents encode semantic features directly without modifying the foundational network parameters. Similarly, our approach to optimizing motion embeddings directly targets temporal features, avoiding the indirect and potentially entangled effects of parameter-level adjustments such as those in Temporal LoRA.
> > > > >
> > > > > We have also addressed your concerns regarding empirical and theoretical distinctions, time and memory comparisons, and clarified phrasing issues (e.g., “differential”). Furthermore, we have provided additional experiments and visualizations to demonstrate the advantages of our approach over Temporal LoRA, VMC, and MotionDirector.
> > > > >
> > > > > Given these updates and the contributions of our method, we kindly ask you to consider raising your score to **6**. Your support would be invaluable in helping us share our findings with the broader research community. Thank you again for your time and insightful feedback.
> > > > >
> > > > >
> > > > > **References**
> > > > > [1] StyleClip: Text-Driven Manipulation of StyleGAN Imagery
> > > > >
> > > > > [2] StyleGAN-NADA: CLIP-Guided Domain Adaptation of Image Generators

---

> > > > > ### Author Response · Authors · 2024-12-03
> > > > >
> > > > > Thank you once again for your thoughtful response and for recognizing our efforts to address your concerns. We sincerely appreciate your engagement.
> > > > >
> > > > > As the discussion deadline approaches, **we kindly ask you to consider raising your score**, as it appears that we have addressed all of your concerns. Your support in this regard would mean a great deal to us and would significantly help in advancing our submission.
> > > > >
> > > > > Once again, we deeply value the time and effort you have dedicated to reviewing our work and contributing to its improvement. Thank you for your consideration!

---

### Official Review · Reviewer_V2wC · 2024-10-29

**Soundness:** 3
**Presentation:** 3
**Contribution:** 3
**Rating:** 6
**Confidence:** 4

**Summary:**

This paper proposes motion embeddings for model control. The main idea is to adjust the temporal attentions with the features extracted from the source guidance video. Specifically, this paper proposes a 1D motion Q-K embedding to add to the Q and K features to guide the global motions. Moreover, a 2D motion V embedding with differencing pre-processing is proposed to add to the V features to guide the local motions. The experimental results show that motion embeddings can be trained to effectively guide the motion of the output video without being influenced by the appearance of the source video.

**Strengths:**

**Originality**: the idea of combining 1D motion Q-K embeddings and 2D motion V embeddings is novel to me. The idea of using 1D to remove the spatial information is not new as in DMT. The idea of using differencing operations is also not new as in DMT. However, combining the two kinds of embeddings for two-level global and local motion control is interesting.

**Quality**: The experimental results generally show that the proposed method is effective and show superior performance over the other state-of-the-art methods in CVPR 24 and ECCV 24. From the videos in the supplementary materials, the performance of the proposed method is satisfying. And the effectiveness of each component has been verified by the ablation study.

**Weaknesses:**

As for **clarify**, though the selection of the 1D and 2D embeddings has been studied in the ablation study, generally speaking, it is still hard to understand why the QK should use 1D embeddings and why the V should use 2D embeddings. There lacks some intuitive explanation on why we should use such settings.

Another **unclear part** is that, 2D embeddings use differencing operations to debias the appearance. However, from Eq. (5), only features with $j>1$ are debiased. When $j=1$, the original features with appearance information are used. Why not debiasing the first feature, for example, differencing $j=1$ with $j=L$?

**Questions:**

**Questions**

1.	Could the authors provide an intuitive explanation on why we should use 1D and 2D for QK and V, respectively?
2.	Why not differencing the first 2D embedding?
3.	Why the frames of the output video are less than the source video. For example, supp\results\teaser\4.mp4 (1 seconds) and 4_source.mp4 (2 seconds)

**Small issues**

Line 191, $F$ should be $\mathbf{F}$

Line 320: `Diffusion Motion Transfer - CVPR24 (DMT) (Yatim et al., 2023), Video Motion Customization - CVPR24 (VMC) (Jeong et al., 2023)`
As DMT and VMC have been accepted to CVPR, the reference should be correspondingly updated to Yatim et al., 2024, Jeong et al., 2024

**Details Of Ethics Concerns:**

N.A.

---

> ### Author Response · Authors · 2024-11-20
>
> Thank you for your insightful feedback! Below, we provide clarifications and address your concerns:
>
> ---
>
> ## **Clarification**
>
> In terms of computation, the motion query and key embeddings are added to the query and key vectors. They directly impact the attention map computation and thus help the model understand frame-to-frame relationships more effectively. These embeddings are well-suited for capturing global relationships, such as camera movements like panning, tilting, zooming, and rotation. These movements tend to affect all spatial regions similarly, making **1D embeddings** an efficient choice for encoding overall frame-to-frame correlations without modeling detailed local variations.
>
> On the other hand, motion value embeddings are added to the value vectors, directly impacting video features during the aggregation process. The **2D embeddings** for V are designed to model localized and complex temporal changes within frames, which is crucial when different spatial regions exhibit distinct movement patterns. For example, scenes with multiple objects moving independently or intricate human actions require a more granular understanding of spatial distinctions. This allows the spatially localized motion information to be effectively preserved and incorporated into the final representation.
>
> ---
>
> ## **Unclear Part**
>
> Thank you for your insightful comment. Based on our empirical observations, we found that debiasing the first feature does not lead to significant differences in performance. To maintain simplicity and clarity in our approach, we opted to keep the current formulation.
>
> The inconsistency in video duration is caused by the frames per second (fps). It can be confirmed that the number of frames in the corresponding videos is the same, but due to different fps, their durations may vary when played back in a video player.

---

> > ### Comment · Reviewer_V2wC · 2024-11-23
> > **Thanks for the response**
> >
> > Most of my questions are answered.
> > However, I agree with Reviewer qixD that the interpretability of the learned embeddings is important.
> > For example, the authors claim the importance of the 2D V embeddings for local movement. \
> > However, in Fig. 7 of the supp html, the video used for ablation study only contains global movement without localized and complex temporal changes, making the response to my question still less clear enough.
> >
> >
> > In addition, the authors are suggested to manually adjust the fps of all video results to make them more easily compared.

---

> > > ### Author Response · Authors · 2024-11-24
> > >
> > > We sincerely appreciate your feedback! Regarding the contribution of 2D V embeddings to local movement, this is addressed in **supp/index.html** under the **Embedding Ablation Section**, where we demonstrated that relying solely on the 1D QK embedding is insufficient to capture the **local movement** in the source video.
> > >
> > > We fully understand your desire to enhance the interpretability of motion representation. However, we hope you can appreciate the inherent difficulty of this task.
> > > Beyond the challenges associated with the latent dimension, it's noteworthy that no reviewers have proposed concrete visualization methods for 2D V embeddings, further underscoring the difficulty of effectively visualizing them. Despite this, we have made a thorough attempt to provide examples in **supp/index.html** under the **Embedding Ablation Section**, which we believe represent a feasible approach.
> > >
> > > That said, we would like to emphasize that these visualization limitations should not detract from the significant contributions of our method, including its improvements in **motion customization** and the valuable insights it offers for **appearance debiasing**.
> > >
> > > Lastly, we have noted your suggestion to manually adjust the FPS of the video results for easier comparison and will ensure this is addressed. Thank you again for your thoughtful and constructive feedback.

---

> > > > ### Comment · Reviewer_V2wC · 2024-11-24
> > > > **About the ablation study**
> > > >
> > > > I saw that two videos in **Embedding Ablation Section** but that experiment has different settings from **Figure 7**.
> > > >
> > > > Could the authors provide the results on these two videos in the same settings as **Figure 7** in **papervideos.html**?
> > > > That's to say, `Source, 1D QK & 1D V, 2D QK & 1D V, 2D QK & 2D V, Ours - 1D QK & 2D V`

---

> > > > > ### Author Response · Authors · 2024-11-24
> > > > >
> > > > > Thank you for your valuable comment. We place the results on these two videos in the same settings as Fig.7 in **structure ablation section** in **index.html** (new updated).
> > > > >
> > > > > ---
> > > > >
> > > > > It's worth noting that for simple camera motion like desert pan-right, most structures yield reasonable result.
> > > > >
> > > > > However, **this is not the case for complicated object movement** like tiger walking. Only the 2d v structure enables the model to produce faithful results.
> > > > > Moreover, when using both 2d qk and 2d v, it becomes challenging to avoid appearance overfitting, even when a *differencing operation* is applied.
> > > > >
> > > > > In conclusion, the combination of 1d qk and 2d v structures proves to be the most effective for motion tasks via extended experiments. The former focuses only on frame interaction, while the latter effectively handles local movement.

---

### Official Review · Reviewer_uLfu · 2024-11-03

**Soundness:** 3
**Presentation:** 4
**Contribution:** 3
**Rating:** 8
**Confidence:** 5

**Summary:**

The paper proposes a motion customization method for video diffusion models. Specifically, it introduces two types of motion embeddings to capture the global relationships between frames and the spatial movements between frames. The former excludes spatial dimensions to effectively disentangle the temporal dynamics from the spatial information. Qualitative and quantitative comparisons, along with ablations, reveal the effectiveness of this method.

**Strengths:**

Method:

The method appears pretty solid, it is logical and intuitive. Disentangling the spatial from the temporal, and learning the temporal dynamics using motion key-query and motion-value embeddings in attention is a good way to approach the problem.

Experiments

There are qualitative results, and quantitative comparisons with various metrics - there could be some changes in this part, elaborated more in the weaknesses.

Paper writing

The paper is well written and easy to understand.

**Weaknesses:**

Experiments

I would really like to see more qualitative results. The results provided in the paper and supplementary are extremely limited, which raises questions on whether the method works well only on a small set of videos.

Additionally, I wonder why there are no comparisons with Tune-A-Video. It'll be great if these comparisons can be provided.

There are also no comparisons with other motion customization methods such as AnimateDiff, MotionMaster, etc - is that because those methods require a specific motion module to be trained?

Method

One clarification - I assume the method is still zero-shot? Are the motion embeddings pre-trained on any videos, or is it all done using just the source video (from scratch)?

**Questions:**

Please answer the questions in the weaknesses.

In addition, I have another (open-ended) question: How dependent is the performance of the method on the backbone foundational model? I understand that the results would naturally improve as stronger backbones are used, but I want to understand if something like Tune-A-Video + strong backbone model would be equivalent to this method + strong backbone model?

Are most of the limitations stated in line 83-92 because of the method or the backbone?

I am happy to revise my score if all my concerns are resolved!

Post rebuttal comment: I have increased my score based on the authors rebuttal.

---

> ### Author Response · Authors · 2024-11-20
>
> Thank you for your insightful feedback! We sincerely appreciate your thoughtful questions and suggestions. Below, we address your concerns and provide clarifications:
>
> ---
>
> ### **Experiments**
> We have included additional qualitative results in the supplementary materials. Kindly refer to the attached file for more details (The *More result* and *Motion Embeddings for ZeroScope and AnimateDiff* sections in newly updated **supp/index.html**).
>
> For quantitative comparisons with Tune-A-Video, please see **Table 1 in the Appendix**, where we present a thorough evaluation. Furthermore, we have included visual examples to highlight the differences between our method and Tune-A-Video for a more intuitive comparison  (The *Compare with Tune-A-Video* section in newly updated supp/index.html).
>
> Regarding **AnimateDiff** and **MotionMaster**, we acknowledge their significance but would like to clarify their objectives and how they differ from ours:
> - **AnimateDiff**: This approach aims to extend image generation models to video generation. In contrast, our method focuses on motion customization in videos generated by video generation models. These objectives are complementary. To emphasize this, we present results combining our method with AnimateDiff in the paper, showcasing their compatibility.
> - **MotionMaster**: This method requires additional segmentation model to separate foreground and background, then estimates motion by the later. Given the foreground motion is calculated by background, it's challanging for this method to accurately stimulate precise object motion, limiting its application in motion related task.
>
> ---
>
> ### **Methods**
> Our approach is designed for **one-shot motion customization**. For each source video, we generate a unique set of motion embeddings entirely from scratch, without any reliance on pre-training on other videos. This ensures that our method achieves diverse and tailored motion effects based on varying text prompts, providing significant flexibility.
>
> ---
>
> ### **Questions**
> We have implemented our method using multiple foundational models, including AnimateDiff, ZeroScope, and VideoCrafterV2, as demonstrated in the paper. While different backbone models influence aspects like text responsiveness and temporal consistency, our approach consistently preserves and customizes the motion from the reference video, showcasing the robustness of our method across diverse settings.
>
> **Tune-A-Video**, on the other hand, focuses on **transforming a text-to-image (T2I) model into a text-to-video (T2V) model** using a single text-video pair. Unlike our method, it does not leverage video generation models as foundational backbones. As a result, its motion fidelity is inherently limited compared to our method, as illustrated in **Table 1 in the Appendix**.
>
> The limitations discussed in **lines 83–92** of our paper refer specifically to the challenges of motion customization itself. These arise from the **incomplete design considerations** of alternative approaches, which introduce inherent methodological flaws, ultimately compromising the quality of the generated results.
>
> ---
>
> We hope these responses clarify your concerns. Thank you once again for your valuable feedback, and we look forward to further engaging discussions!

---

> > ### Author Response · Authors · 2024-11-24
> >
> > Dear Reviewer,
> >
> > Thank you again for your valuable feedback and thoughtful comments during the discussion phase. We would like to kindly remind you that the discussion period will conclude on **November 26th**. If you have any additional questions, concerns, or clarifications you would like us to address, we would be more than happy to provide prompt responses.
> >
> > Your insights have been instrumental in shaping the final version of our submission, and we greatly appreciate your time and effort in engaging with our work.
> >
> > Thank you for your attention, and we look forward to hearing from you!

---

> > > ### Comment · Reviewer_uLfu · 2024-11-24
> > >
> > > Thanks to the authors for their efforts! I am satisfied with the responses, and have increased my score from 6 to 8.

---

### Author Response · Authors · 2024-12-03

Dear reviewers,

Thank you for your detailed and thoughtful feedback. We deeply value your constructive critiques and have addressed them to the best of our ability through comprehensive experiments and clarifications. As the discussion deadline approaches, we kindly ask you to consider whether our responses meet your expectations.

Thank you once again for your engagement.

Best,

Authors.

---

### Meta-Review · Area_Chair_av9N · 2024-12-21

**Metareview:**

The paper addresses the problem of motion customization for video models (given a video, generate a video with its approximate motion) and proposes a "motion embedding" which is designed and trained specifically to capture the motion in the source video. Specifically, this motion embedding consists of motion query-key embeddings which influence the temporal attention query-keys and motion value embeddings which influence the temporal attention values. To ensure that this embedding does not entangle appearance and motion, they design the motion query-key to be 1D (to avoid any spatial knowledge) and specifically debias the motion value. The paper demonstrates strong quantitative results in text similarity, motion fideltiy, temporal consistency, FID, and user preference. In my opinion, due to the small number of samples, it is difficult to fully evaluate the qualitative results (particularly wrt Motion Director).

The main strength of the paper is the relatively clear presentation and strong quantitative results. The core weakness as identified by both Reviewers qixD and 5Vdk is the very close similarity of the method with MotionDirector and the lack of clarity as to the primary differences between the two papers. Since it was published in ECCV24 MotionDirector is prior work and should be treated as such.

The paper demonstrates superior quantitative results over MotionDirector though due to the small number of examples, I found it hard to qualitatively evaluate the two in standard setting. It is however worth noting that this improvement is more pronounced when considering the MSE loss (as opposed to the hybrid).

I read through both this paper, Motion Director, and the extensive discussion in the review section between the author and reviewers. The core concern of Reviewers qixD and 5Vdk is the similarity of a LoRA to that of an embedding which acts as an additive edit to the attention weights. While there are some differences in the exact formulation, the main argument from the authors that I found most convincing is that their method could work on full attention models whereas the LoRA is dependent on the separation between temporal and spatial attention. However, as far as I understand, this is just an argument the authors used and not empirically demonstrated. I checked the unet_3d_condition definition in the diffusers library (used by Zeroscope_v2) and it seems to be a TransformerTemporal which does factorized space-time attention. While I believe that the embedding formulation should work with a full attention model, I think this needs to be empirically demonstrated. In addition, the paper needs to be rewritten to draw a much closer comparison, especially technically, with Motion Director. I think one very simple way that this paper could be accepted in future conference is to demonstrate that their method works on full attention models – then there will be no ambiguity between theirs and MotionDirector. I'm generally sympathetic to the author's argument that their improvement performance demonstrates that there is a difference. However, in the current version of the paper, as both Reviewers qixD, 5Vdk, and I struggled with, it is very hard to understand what the difference is. In order to be accepted, this difference needs to be directly explained.

While I think it's a very close call, I agree with Reviewers qixD and 5Vdk and would rate this paper slightly below, though I strongly encourage the authors to add detail to the paper explaining the difference between this and MotionDirector and then demonstrating this difference empirically on for example a full attention model. I would also encourage the authors to include more qualitative samples in the future, particularly for video papers.

**Additional Comments On Reviewer Discussion:**

Both Reviewers qixD and 5Vdk focus primarily on the comparison with MotionDirector and give a rating of 5 (marginally below) due to the technical similarity and lack of clarity. I agree with their concern about the differences between this paper and MotionDirector. More detail above.

Reviewer uLfu gave this paper an 8 (accept, good paper) and said that they was willing to champion this paper. However, when asked about the comparison with MotionDirector, they quoted the author's response to their questioning which notably does not mention MotionDirector. The reviewer asked about AnimateDiff, which is a text2video technique, and MotionMaster which does explicit segmentation and motion estimation. Neither of these two papers in the quote are MotionDirector. Since the reviewer did not adequately consider the main concern (the comparison with MotionDirector) this review was weighed less than the others.

Reviewer V2wC gave this paper a 6 and in their review focused primarily on the interpretability of the embedding as well as clarifying some details.

---

### Decision · Program_Chairs · 2025-01-22

Reject